# BAYESIAN NEURAL NETWORK PARAMETERS PROVIDE INSIGHTS INTO THE EARTHQUAKE RUPTURE PHYSICS

## ABSTRACT

I present a simple but informative approach to gain insight into the Bayesian neural network (BNN) trained parameters. I used 2000 dynamic rupture simulations to train a BNN model to predict if an earthquake can break through a simple 2D fault. In each simulation, fault geometry, stress conditions, and friction parameters vary. The trained BNN parameters show that the network learns the physics of earthquake rupture. Neurons with high positive weights contribute to the earthquake rupture and vice versa. The results show that the stress condition of the fault plays a critical role in determining its strength. The stress is also the top source of uncertainty, followed by the dynamic friction coefficient. When stress and friction drop of a fault have higher value and are combined with higher weighted neurons, the prediction score increases, thus fault likely to be ruptured. Fault's width and height have the least amount of uncertainty, which may not be correct in a real scenario. The study shows that the potentiality of BNN that provides data patterns about rupture physics to make an additional information source for scientists studying the earthquake rupture.

## 1 INTRODUCTION

Because of the limited observational data and computational cost, geoscientists often rely on simple low-resolution simulations to study physical systems such as dynamic earthquake rupture, long-term tectonic process, etc. Such simplified models are indeed a powerful tool beside the observational data but sometimes cannot capture the proper physics of the system. As a result, it becomes difficult to accurately identify and understand the underlying causes.

Machine learning (ML) approaches have been successfully used to solve many such geophysical problems with limited data and require computational overhead. For example, Ahamed & Daub (2019) used neural network and random forest algorithms to predict if an earthquake can break through a fault with geometric heterogeneity. The authors used 1600 simulated rupture data points to train the models. They identified several patterns responsible for earthquake rupture.

Machine learning approaches are also used in seismic event detection (Rouet-Leduc et al., 2017), earthquake detection (Perol et al., 2018), identifying faults from unprocessed raw seismic data (Last et al., 2016) and to predict broadband earthquake ground motions from 3D physics-based numerical simulations (Paolucci et al., 2018). All the examples show the potential application ML to solve many unsolved geophysical problems.

However, the machine learning model's performance usually depends on the quality and quantity of data. Bad quality or insufficient data increases the uncertainty of the predictions (Hoeting et al., 1999; Blei et al., 2017; Gal et al., 2017). Therefore, estimating the source of uncertainty is vital to understanding the physics of earthquake rupture and seismic risk. On top of that black-box nature of the ML algorithms inhibits mapping the input features with model output prediction. As a result, it becomes challenging for scientists to make actionable decisions.

To overcome insufficient earthquake rupture data, I used the Bayesian neural network algorithm to develop a model reusing the simulations data of Ahamed & Daub (2019). I present an exciting approach to learning the patterns of earthquake ruptures from the trained model parameters. Unlike regular neural networks, BNN works better with a small amount of data and provides prediction uncertainty. The approach gives more information on rupture physics than the traditional geophys-

ical methods. I also describe the workflow of (1) developing a BNN and (2) estimating prediction uncertainty.

## 2 EARTHQUAKE RUPTURE SIMULATIONS

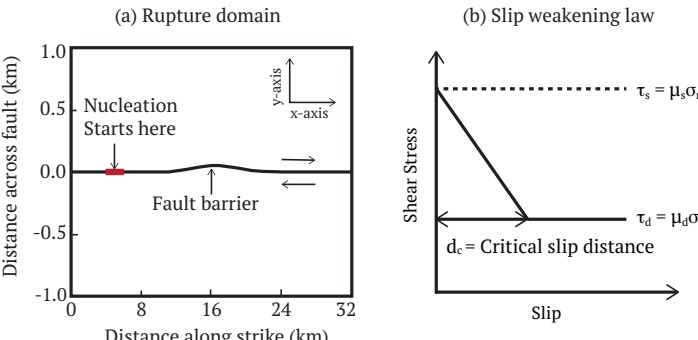

Figure 1: (a) A zoomed view of the two-dimensional fault geometry. The domain is 32 km long along the strike of the fault and 24 kilometers wide across the fault. The rupture starts to nucleate 10 km to the left of the barrier and propagates from the hypocenter towards the barrier, (b) Linear slip-weakening friction law for an earthquake fault. The fault begins to slip when the shear stress reaches or exceeds the peak strength of $\tau_s$. $\tau_s$ decreases linearly with slip to a constant dynamic friction $\tau_d$ over critical slip distance ($d_c$). The shear strength is linearly proportional to the normal stress $\sigma_n$, and the friction coefficient varies with slip between $\mu_s$ and $\mu_d$.

I used the simulated earthquake rupture dataset created by Ahamed & Daub (2019). The simulations are a two-dimensional rupture, illustrated in figure. 1. The domain is 32 km long and 24 km wide. Figure 1a shows the zoomed view of the original domain for better visualization of the fault barrier. Rupture is nucleated 10 km to the left of the barrier and propagates towards the barrier. In each simulation, eight parameters were varied: x and y components of normal stress (sxx and syy), shear stress (sxy), dynamic friction coefficient, friction drop ($\mu_s - \mu_d$), critical slip distance ($d_c$), and width and height of the fault. The fault starts to break when the shear stress ($\tau$) on the fault exceeds the peak strength $\tau_s = \mu_s\sigma_n$, where $\mu_s$ and $\sigma_n$ are the static friction coefficient and normal stress, respectively. Over a critical slip distance $d_c$, the friction coefficient reduces linearly to constant dynamic friction $\mu_d$.

1600 simulation data points were used to train, and 400 were used to test the model performance. The training dataset has an imbalance class proportion of rupture arrest (65%) and rupture propagation (35%). To avoid a bias toward rupture arrest, I upsampled the rupture propagation examples. Before training, all the data were normalized by subtracting the mean and dividing by the standard deviation.

## 3 BAYESIAN NEURAL NETWORK

In a traditional neural network, weights are assigned as a single value or point estimate. In a BNN, weights are considered as a probability distribution. These probability distributions are used to estimate the uncertainty in weights and predictions. Figure 2 shows a schematic diagram of a BNN where weights are normally distributed. The posterior network parameters are calculated using the following equation:

$$P(W|\mathbf{X}) = \frac{P(\mathbf{X}|W)P(W)}{P(\mathbf{X})} \tag{1}$$

Where $\mathbf{X}$ is the data, $P(\mathbf{X}|W)$ is the likelihood of observing $\mathbf{X}$, given weights ($W$). $P(W)$ is the prior belief of the weights, and the denominator $P(\mathbf{X})$ is the probability of data which is also known as evidence. The equation requires integrating over all possible values of the weights as:

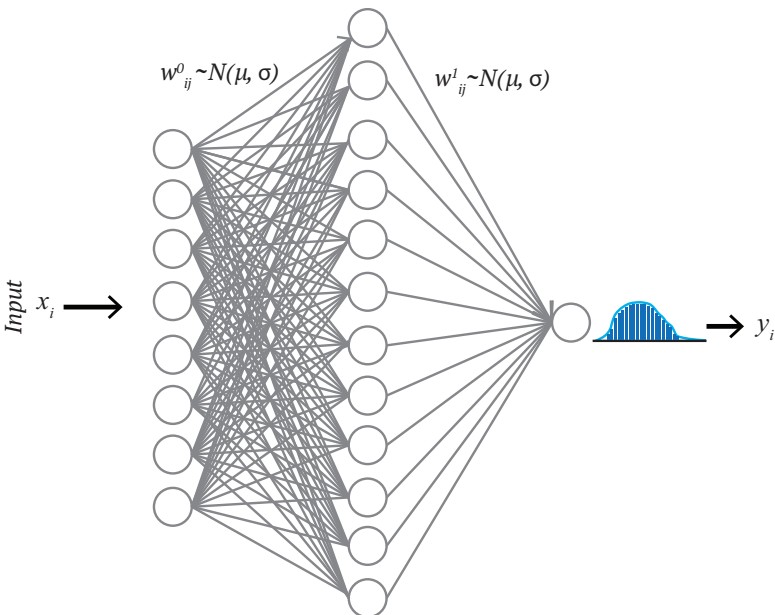

Figure 2: The schematic diagram shows the architecture of the Bayesian neural network used in this work. The network has one input layer with eight parameters, one hidden layer with twelve nodes, and an output layer with a single node. Weights between input and hidden layers are defined by $w_{ij}^0$, which are normally distributed. $i, j$ are the node input and hidden layer node index. Similarly, $w_{jk}^1$ is the normal distribution of weights between the hidden and the output layer. $\mu$ and $\sigma$ are the mean and standard deviation. At the output node, the network produces a distribution of prediction scores between 0 and 1.

$$P(\mathbf{X}) = \int P(\mathbf{X}|W)P(W)dW. \tag{2}$$

Integrating over the indefinite weights in evidence makes it hard to find a closed-form analytical solution. As a result, simulation or numerical based alternative approaches such as Monte Carlo Markov chain (MCMC) and variational inference(VI) are considered. MCMC sampling is an inference method in modern Bayesian statistics, perhaps widely studied and applied in many situations. However, the technique is slow for large datasets and complex models. Variational inference (VI), on the other hand, is faster. It has been applied to solve many large-scale computationally expensive neuroscience and computer vision problems (Blei et al., 2017).

In VI, ar new distribution $Q(W|\theta)$ is considered that approximates the true posterior $P(W|\mathbf{X})$. $Q(W|\theta)$ is parameterized by $\theta$ over $W$ and VI finds the right set of $\theta$ that minimizes the divergence of two distributions through optimization:

$$Q^*(W) = \underset{\theta}{\operatorname{argmin}} \, \mathbf{KL}\left[Q(W|\theta)||P(W|\mathbf{X})\right] \tag{3}$$

In equation-3, $\mathbf{KL}$ or Kullback–Leibler divergence is a non-symmetric and information theoretic measure of similarity (relative entropy) between true and approximated distributions (Kullback, 1997). The KL-divergence between $Q(W|\theta)$ and $P(W|\mathbf{X})$ is defined as:

$$\mathbf{KL}\left[Q(W|\theta)||P(W|\mathbf{X})\right] = \int Q(W|\theta) \log \frac{Q(W|\theta)}{P(W|\mathbf{X})} dW \tag{4}$$

Replacing $P(W|\mathbf{X})$ using equation-1 we get:

$$\mathbf{KL}\left[Q(W|\theta)||P(W|\mathbf{X})\right] = \int Q(W|\theta) \log \frac{Q(W|\theta)P(\mathbf{X})}{P(\mathbf{X}|W)P(W)} dW$$

$$= \int Q(W|\theta) \left[\log Q(W|\theta)P(\mathbf{X}) - \log P(\mathbf{X}|W)P(W)\right] dW$$

$$= \int Q(W|\theta) \log \frac{Q(W|\theta)}{P(W)} dW + \int Q(W|\theta) \log P(\mathbf{X}) dW$$

$$- \int Q(W|\theta) \log P(\mathbf{X}|W) dW \tag{5}$$

Taking the expectation with respect to $Q(W|\theta)$, we get:

$$\mathbf{KL}\left[Q(W|\theta)||P(W|\mathbf{X})\right] = \mathbb{E}\left[\log \frac{Q(W|\theta)}{P(W)}\right] + \log P(\mathbf{X}) - \mathbb{E}\left[\log P(\mathbf{X}|W)\right] \tag{6}$$

The above equation shows the dependency of $\log P(\mathbf{X})$ that makes it difficult to compute. An alternative objective function is therefore, derived by adding $\log P(\mathbf{X})$ with negative KL divergence. $\log P(\mathbf{X})$ is a constant with respect to $Q(W|\theta)$. The new function is called as the evidence of lower bound (ELBO) and expressed as:

$$ELBO(Q) = \mathbb{E}\left[\log P(\mathbf{X}|W)\right] - \mathbb{E}\left[\log \frac{Q(W|\theta)}{P(W)}\right] \tag{7}$$

$$= \mathbb{E}\left[\log P(\mathbf{X}|W)\right] - KL\left[Q(W|\theta)||P(W|\mathbf{X})\right] \tag{8}$$

The first term is called likelihood, and the second term is the negative KL divergence between a variational distribution and prior weight distribution. Therefore, ELBO balances between the likelihood and the prior. The ELBO objective function can be optimized to minimize the KL divergence using different optimizing algorithms like gradient descent.

## 4 TRAINING BAYESIAN NEURAL NETWORK

The BNN has the same NN architecture used in Ahamed & Daub (2019) to compare the performance between them. Like NN, BNN has one input layer with eight parameters, one hidden layer with twelve nodes, and one output layer (Figure 2). A nonlinear activation function `ReLu` (Hahnloser et al., 2000) was used at the hidden layer. `ReLu` passes all the values greater than zero and sets the negative output to zero. The output layer uses `sigmoid` activation function, which converts the outputs between zero and one.

Prior weights and biases are normally distributed with zero mean and one standard deviation. Figure 3 shows the log density of prior and posterior weights ($w_{ij}^k$) and biases ($b_j^k$). $i$ and $j$ are the index of the input and hidden layer nodes. $i$ ranges from 0 to 7, and $j$ ranges from 0 to 11. $k$ is the index that maps two layers. For example, $w_{15}^0$ is the weight between the first input node and the fifth hidden node. The output node of the last layer produces a distribution of prediction scores between 0 and 1. The prediction distributions are used to compute standard deviation, which is the uncertainty metric.

Adam optimization (extension of stochastic gradient descent) was used to minimize the KL divergence by finding a suitable variational parameter $\theta$. The initial learning rate is 0.5, which exponentially decays as the training progresses. To train the BNN, I use `Edward` (Tran et al., 2016; 2017), `TensoFlow` (Abadi et al., 2015) and `Scikit-learn` (Pedregosa et al., 2011). `Edward` is a Python-based Bayesian deep learning library for probabilistic modeling, inference, and criticism. All the training data, codes, and the corresponding visualizations can be found on the Github repository: `https://github.com/msahamed/earthquake_physics_bayesian_nn`

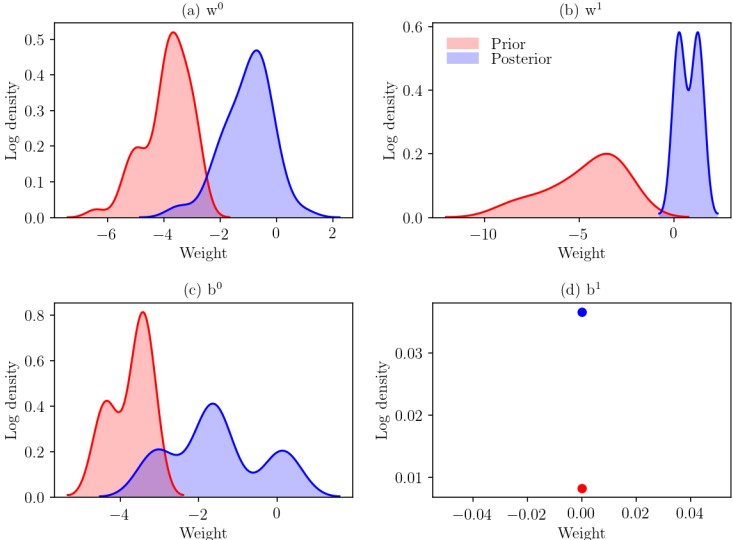

Figure 3: The graph shows the distribution of prior and posterior mean weights (a) $w^0$ (b) $w^1$ and biases (c) $b^0$ (d) $b^1$. Both location of the mean and magnitude of density of the posterior distributions (weights and biases) are noticeably different from the priors which indicates that BNN has learned from the data and adjusted the posterior accordingly.

## 5 PRIOR AND POSTERIOR PARAMETER DISTRIBUTION

To evaluate the parameters (weights and biases) of the BNN, 1000 posterior samples of $w_{ij}^0$, $w_{jk}^1$, $b^0$ and $b^1$ were used. Figure 3 shows the prior and posterior distribution of mean weight and biases. The posterior location of the mean and density of the weights and biases are different from their priors. For example, the location of $w^0$ shifts toward non-negative value, while the density remains similar. Whereas, the $w^1$, $b^0$, and $b^1$ have a different posterior mean location and density than their prior. The differences between prior and posterior indicate that the BNN has learned from the data and adjusted the posterior distribution accordingly.

The performance of the BNN was evaluated using 400 test simulations. For a given test example, 1000 posterior samples were used to determine the class and associated uncertainty. Uncertainty is the standard deviation of the prediction scores. The test accuracy of the BNN is $83.34\%$, which is $2.34\%$ higher than NN. In the following subsection, I discuss how uncertainty can help us understand physics and find the parameter combinations responsible for an earthquake rupture.

### 5.1 LEARNING FROM NETWORK PARAMTERS

Estimating the uncertainty of neural network parameters (weights) helps us understand the black box behavior. The illustration in Figure 4 shows the mean and standard deviation of $W^0$ and $W^1$. $W^0$ maps the inputs to the hidden layer nodes, whereas $W^1$ maps the output node of the output layer to the hidden layer's nodes. The colors in each cell indicate the magnitude of a weight that connects two nodes. In the input and hidden layer, the ReLu activation function was used. ReLu passes all the positive output while setting a negative output to zero. Sigmoid activation is used at the output layer, which pushes the larger weights toward one and smaller or negative weights toward zero. Therefore, positive and high magnitude weights contribute to the earthquake rupture and vice versa.

In $w^0$, nodes connected to friction drop, dynamic friction, shear, and normal stresses have variable positive and negative weights. The corresponding node in $w^1$ also has a strong positive or negative magnitude. For example, node-12 has both positive and negative weights in $w^0$, and the corresponding node in $w^1$ has a high positive weight. Similarly, node-4 has a substantial negative weight in $w^1$, and the corresponding nodes in $w^0$ have both positive and negative weights. On the other hand,

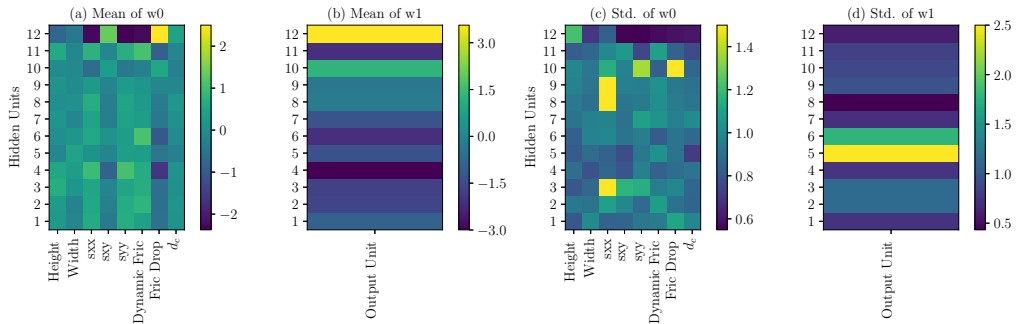

Figure 4: The illustration shows the posterior mean and standard deviations of $w^0$ and $w^1$. (a) $w^0$ that map the inputs to the nodes of the hidden layer. The eight input parameters are on the horizontal axis, and the twelve nodes are on the vertical axis. The colors in each cell are the magnitudes of mean weight. (b) $w^1$ maps the hidden layer to the output layer. (c) The standard deviation of $w^0$. Shear stress connected to node-4 of the hidden layer has the highest uncertainty. Similarly, the weights associated with the input parameters and node-5 have high uncertainty. Whereas, the weights associated with the input parameters and the nodes 7-11 have relatively low uncertainty. (d) Uncertainty of the weights associated with the hidden layer nodes and the output node. Weights in node 7 and 8 have high uncertainty while the rest of the weights have relatively low uncertainty.

width, height, and $d_c$ have a similar magnitude of weights, and the corresponding nodes in $w^1$ have a moderate magnitude of weights. Thus the variable weights make friction drop, dynamic friction, shear, and normal stresses influential on the prediction score. Therefore, we can now detect the important features and their uncertainties from any input combinations.

For example, node-10 of $w^1$ has positive weight (Figure 4b). In $w^0$, the corresponding connecting input features, friction drop, and shear stress have positive weight and low uncertainty, whereas the rest have a similar magnitude of weight. The combination of high friction drop and shear stress weight and low weights of other features increase the prediction score, thus likely to cause rupture to propagate. Friction drop and normal stresses also have high uncertainty (Figure 4c). For this combination of patterns, friction drop and normal stresses influence the prediction strongly and are also the sources of uncertainty. Thus, it gives us the ability to investigate any rupture propagation example in terms of uncertainty.

## 5.2 Prediction Uncertainity

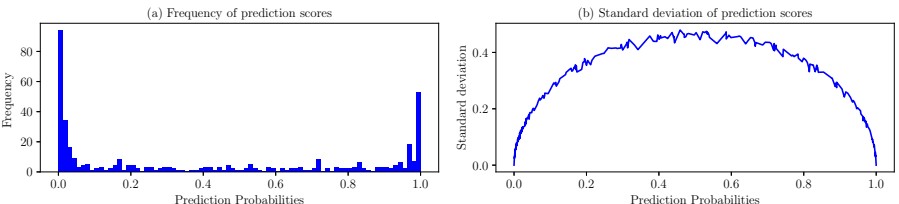

Figure 5: The graph shows (a) frequency and (b) standard deviation of posterior prediction scores of the test data. Prediction scores are skewed toward the left side while slightly less on the right. The observation is consistent with the proportion of the rupture arrest (272) and rupture propagation (120) in the test data. Prediction scores close to zero are related to rupture arrest, and scores around one are the rupture propagation. Standard deviations are high with scores around 0.5.

Figure 5a and b show the frequency and standard deviation of test data prediction scores. The distribution is more skewed toward smaller scores than the higher ones. Scores close to zero are associated with rupture arrest, whereas scores close to one are rupture propagations. The observation is consistent with the class proportion of the test data. Rupture arrest has a higher number of

examples (272) than rupture propagation (120). Figure 5a shows that scores roughly between 0.35 and 0.75 have a fewer number of examples and thus high uncertainty. The likely reason for the high uncertainty is that the examples have both rupture propagation and arrest properties. As a result, the model gets confused and cannot classify the example correctly; thus, the misclassification rate is high in this region. The network's performance could be improved if sufficient similar example data are added to the training dataset.

## 5.3 ROLE OF INPUT FEATURES

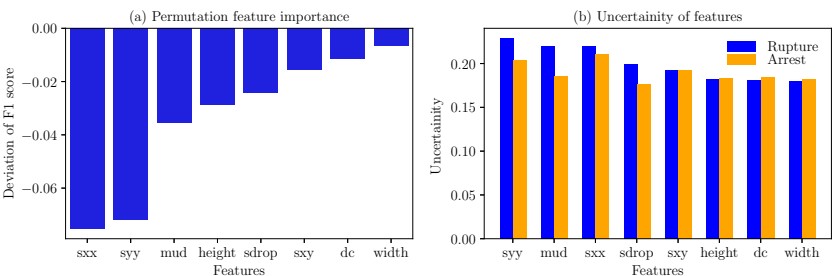

Figure 6: The illustration shows (a) permutation feature importance and (b) their uncertanitities.

I used the permutation importance method to determine the source of uncertainty in the test data. Permutation importance is a model agnostic method that measures a feature's influencing capacity by shuffling it and measuring corresponding global performance deviation. If a feature is a good predictor, altering its values reduces the model's global performance significantly. The shuffled feature with the highest performance deviation is the most important and vice versa. In this work, the F-1 score is the performance measuring metric.

Figure 6(a) shows the permutation importance of all the features. Normal stresses (sxx and syy) have the highest F-1 score deviation, which is approximate 7% less than the base performance, followed by the dynamic friction coefficient. Geometric feature width has the least contribution role to determine the earthquake rupture. These observations are consistent with the observation of Ahamed & Daub (2019), where the authors rank the features based on the random forest feature importance algorithm.

From the distribution of prediction scores, the standard deviation was calculated for each shuffled feature. Figure 6(b) shows the uncertainty of each feature of earthquake rupture propagation and arrest. All features have higher uncertainty in the rupture propagation compare to the rupture arrest. In both classes, the major portion of uncertainty comes from normal stresses. Shear stress, height, width, and critical distance ($d_c$) have a similar amount of uncertainty in both classes. The dynamic friction coefficient and friction drop are also the comparable uncertainty sources in rupture to propagate while slightly less in earthquake arrest.

The above observations imply that normal stress and friction parameters have a more considerable influence in determining the earthquake rupture. Although the height and width of a fault are not a significant source of uncertainty, they play a more influential role in influencing other features. For example, in a complex rough fault, the variation of the bending angle of barriers affects stress perturbation, consequently increasing the uncertainty. If the angle near the bend is sharp, the traction variation at the releasing and restraining bend is more prominent. On the other hand, if the barrier is broad, the stress perturbation at the restraining and releasing bend is less noticeable. Chester & Chester (2000) found a similar observation that fault geometry impacts the orientation and magnitude of principal stress.

## 6 DISCUSSION AND CONCLUSION

In recent years, deep learning has been used to solve many real-life problems and achieve a state of the art performance. Some examples are facial recognition, language translation, self-driving cars,

disease identification, and more. Such successful applications require millions of data points to train the model and achieve a state of the art performance. However, in many situations, data is limited data like earthquake rupture.

In this work, I used a Bayesian neural network to overcome the small data problem and estimate prediction uncertainty. Rupture simulation created by Ahamed & Daub (2019) was used to develop a model to predict if an earthquake can break a fault. Each 2D simulation fault has a Gaussian geometric heterogeneity at the center. The eight fault parameters of normal shear stress, height, and width of the fault, stress drop, dynamic friction coefficient, critical distance were varied in each of the simulations. Sixteen hundred simulations were used to train the BNN, and 400 were used to test the model's performance.

The test F1 score is 0.8334, which is 2.34% higher than the NN. All the features have a higher uncertainty in rupture propagation than the rupture arrest. The highest sources of uncertainty came from normal stresses, followed by a dynamic friction coefficient. Therefore, these features have a higher influencing capacity in determining the prediction score. Shear stress has a moderate role, but the geometric features such as the fault's width and height are least significant in determining rupture. Test examples with prediction scores around 0.5 have a higher uncertainty than those with low (0-0.30) and high (0.7-1.0) prediction scores. Cases with prediction scores around 0.5 have mixed properties of rupture propagation and arrest.

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
