# OpenReview forum: "Bayesian neural network parameters provide insights into the earthquake rupture physics."
_ICLR.cc/2021/Conference — Reject_

### Official Review · AnonReviewer3 · 2020-10-23
**Interesting subject and paper but may not be enough for ICLR**

**Rating:** 6
**Confidence:** 4

**Review:**

This paper proposes a Bayesian neural network for predicting if an earthquake will break a fault or not, overcoming 'small data problem' and predicting model uncertainty. The data is composed of 8 features and a binary output, and the samples are all coming from simulations. An analysis on the means and standard deviations of the first and last layer of the neural network's weights has been carried out.

The problem is interesting, and the method is useful as the study on the weights means and stds is interesting. Yet, I think the paper is not well polished (many incoherence in the text and the figures) and I don't really understand why a synthetic dataset, coming from physical model equations, needs to go into this complex uncertainty quantification: we are making here a 'meta-model' of something that is fully described by equation, so why using machine learning? I also don't understand why the author talks about a 'small data problem', as here we could simply increase the number of simulated samples? Yet, I can see the interest of such a technique if the goal was to later try to apply it to real data, where some physics might be unknown or too complex. I also don't know if the findings of the paper are of interest for the ICLR community, but as I come from an interdisciplinary field too, I know how hard it can be to find an appropriate and yet good place to publish.

Other questions/Remarks:
 - Equ. 8 : I don't see how we can go from Eq. 6 to Eq. 8? this part is not clear.
 - Please cite the reference papers for the Bayesian NN as well as for ELBO. There is clearly a part missing in the state-of-the-art regarding this, while the equations (such as the long and useless (5) one) are just explaining what is already known in this literature.
 - Figure 4: 'Shear stress connected to node-4 of the hidden layer has the highest uncertainty. Similarly,
the weights associated with the input parameters and node-5 have high uncertainty. Whereas, the
weights associated with the input parameters and the nodes 7-11 have relatively low uncertainty. (d)
Uncertainty of the weights associated with the hidden layer nodes and the output node. Weights in
node 7 and 8 have high uncertainty while the rest of the weights have relatively low uncertainty.'
	--> please update the text or the figures so the numbers match. Same thing for the text. For now, I cannot understand anything.
- Figure 1: Tau_s decreases linearly: not, tau_s is fixed, maybe you meant tau or 'the shear stress'

typos:
- 'perhaps widely studied and applied in many situations' --> 'perhaps'? never seen this word in a paper...
- ar new distribution
- is therefore, derived by
- which is approximate 7% less
- data is limited data

---

### Official Review · AnonReviewer4 · 2020-10-26
**The paper study an interesting topic and application of machine learning in Geophysics. However, the paper do not provide any novel contribution related to machine learning.**

**Rating:** 4
**Confidence:** 5

**Review:**

The manuscript do not provide enough details about  the physics about which the BNN provides insight.
The manuscript employs the well developed machine learning algorithms in an application in Geoscience and do not provide a novel learning algorithm or contribute to machine learning topics. This paper does not meet the  ICLR standards and therefore I can not recommend for publication.

---

### Official Review · AnonReviewer1 · 2020-10-26
**Minor improvements from ANN to BNNs - application new but additional insight of Bayesian methods not clearly provided**

**Rating:** 4
**Confidence:** 4

**Review:**


This paper extends on previous works by Ahamed & Daub (2019), from a two-layers MLP, two a Bayesian NN version of it, for predictiong wether a piece of material will rupture, under some conditions.
Although this increase in model complexity improves performance a little bit, it does not represent a major advance.
The main point of the paper is to show that Bayesian NN allow to get not only a prediciton, but also naturally provide uncertainties on these predictions.
Although the paper clearly explains the basics of BNNs, it does not provide any new insight into them. The application to rupture physics is interesting, but does not seem groundbreaking.

For these reasons, I lean on **rejecting the paper.**

Also, given the github repo referred to, there is a breach of anonymity.

On the paper itself, I have a couple of remarks:
- it is unclear as to where the data comes from. A simulation is mentionned, but not how it works. I see in Ahamed & Daub (2019) that it is a finite element simulation.
- still about the simulation, it is uncelar whether the stress state is heterogeneous or not. It seems it ought to be, however in that case, the description of the stress state would consist in a full field of values, and not just a couple of numerical values.
- the discussions of the uncertainty in various input variable is tedious and does not really highligh how BNNs help to get transparent interpretations.
- Fig 5b: there is a clear sqrt(x)sqrt(1-x) shape of this curve. Do you have an epxlanation for that (and can you check the fit?)
- the conclusion repeats some parts that were stated rearlier. It should instead focus on how BNNs help understand the physics. The physics of rupture itself is not very interesting to a ML audience.

---

### Official Review · AnonReviewer2 · 2020-10-27
**Review on the usage of BNNs to gain insights on the physical procedure of earthquake ruptures through the study of the parameters**

**Rating:** 4
**Confidence:** 3

**Review:**

### Summary
In the present paper, the author intends to get further insights into the physics behind earthquake ruptures using a BNN to model simulated data from the literature. By using a BNN, the parameters of the model are not deterministic scalar values, but complete probability distributions. Studying the change of the distributions in the parameters before and after training, the author tries to extract information about the relative importance of the input variables, and also comprehend the physical mechanisms behind earthquake ruptures. Results are shown in figure 3, on which the change of behavior of the distributions of the parameters can be observed, as well as in figure 4 where the mean and standard deviations for all the parameters are presented. The pattern in figure 6 seems to indicate that variables previously thought to be important in the task of predicting the presence of the rupture, such as normal stress and friction, are also pointed out as being important in this case.  Finally, the authors also claim an improvement in the F1 metric in comparison to previous NN methods.

##### Pros
* The idea of the paper seems well directed, i.e., gaining insight on complex physical procedures using an approach that results in the combination of NNs and a Bayesian approach.
* Using a Bayesian approach is a good way of dealing with small datasets, and also allows to account for the uncertainty of all the latent parameters, while also providing more robust and sensible predictions when new data is presented
* The approach seems to provide results consistent with the literature findings regarding the important variables in the prediction
* The final performance of the algorithm seems to improve on the previous state-of-the-art methods by taking advantage of the properties that the Bayesian approach offers

##### Cons
* The key concern with this paper is that NNs, as well as BNNs, are notoriously black-box algorithms with no easy way of interpreting the inner parameters in most cases. Taking this into consideration, I would suggest the author to motivate in a stronger manner why the usage of BNNs is desirable for the proposed problem, and why not use other already established Bayesian approaches to assess the importance of the input variables.
* Taking into account the previous point, I consider there is a general lack of rigorous experiments that could, in principle, suggest a clear advantage of using BNNs instead of any other approaches. No systematic comparisons with previous methods are present, such as for example with the Random Forest Feature Importance algorithm, which is mentioned a couple of times. If the main goal is to gain insights on the main variables involved in the presence of an earthquake rupture, I would expect a more detailed analysis comparing how good these insights provided by BNNs are and how do they stand in comparison with the established literature.
* Other basic techniques for assessing the importance of the variables in the prediction tasks are not mentioned, although it would be nice to use them as a baseline to compare against. Examples such as PCA, LOO cross-validation and others could be used here.
* The claim of an improvement of 2.34% w.r.t. NNs is not strongly addressed, since the NN experiments are not included here or, at least, there is no mention of the setup of these NNs. As before, there is no systematic comparison between the BNNs trained and the NNs that are used as baselines.
* There is a lengthy discussion on how to obtain the ELBO for VI. However, in the end, there is no final expression for the loss function which is going to be employed. I would appreciate in section 4 an explicit description of the objective of the system since there's no mention of the final binary classification problem anywhere.
* The prediction uncertainties lack a systematic evaluation as well since all that is provided is presented in figure 5. How well do the predictions provided stand against other methods for obtaining final predictive distributions?
* VI is a method whose performance and final predictions are constrained due to its formulation. Is there any reason why using VI instead of any other approach to BNNs? In case that we wanted to study the final predictive distributions, why not use HMC or other, more flexible approaches than VI?
* At the end of the section 5.1, first paragraph, it is claimed that "positive and high magnitude weights contribute to the earthquake rupture and vice versa". This sentence seems a bit confusing since it seems to imply a causal relation between the high magnitude of the weights and the appearing of ruptures. This, I think, is the other way around: very clear rupture conditions imply positive high magnitude weights, which, in turn, return a higher predicted probability of rupture.

##### Minor comments:
* Even though the paper tackles physical phenomena such as earthquake rupture, it does not provide any description of such process or the variables involved. Concepts such as "nucleation" and "fault barrier" should be at least briefly introduced, as well as the "slip weakening law" or the "critical slip distance". A short description of these terms and their relevance to the problem would help to interpret the final results obtained. Also, explicit expressions for the rupture physics would help a lot in section 2 to understand the different roles of the variables and their relations.
* Throughout the whole text it is used the first person while writing. In case there is only one author this can be okay, I only point it out since it seems to be a uncommon choice.
* There are a lot of typos all through the paper! Please perform a careful reading and correct them.
* The description on figure 4 is confusing, does not seem to correspond to the presented images (either that or the text is unclear when selecting the important parts of the figures for the nodes mentioned).
* 4th paragraph of introduction - not all ML algorithms are black boxes! NN are, but other such as linear regression, decision trees, etc., can be very interpretable!
* 5th paragraph of introduction - "exciting" - avoid usage of these type of subjective adjectives all through the paper.
* 5th paragraph of introduction - BNNs may work better with fewer data, but we have to pay close attention to the prior formulation to not introduce unreasonable biases.

---

### Decision · Program_Chairs · 2021-01-07
**Final Decision**

**Decision:**

Reject

**Comment:**

The paper considers an interesting application of Bayesian neural nets to the geophysics domain;  however, the paper does not make a novel contribution from the machine learning perspective, and the improvements on top of the previously proposed approach by Ahamed & Daub (2019) seem to be quite modest. Overall, the paper does not seem to be ready for publication at ICLR.